# Distribution of Small Ruminant Lentivirus Genotypes A and B in Goat and Sheep Production Units in Mexico

**DOI:** 10.3390/vetsci12030204

**Published:** 2025-03-01

**Authors:** Jazmín De la Luz-Armendáriz, Aldo Bruno Alberti-Navarro, Erika Georgina Hernández-Rojas, Andrés Ernesto Ducoing-Watty, Alberto Jorge Galindo-Barboza, José Francisco Rivera-Benítez

**Affiliations:** 1Departamento de Medicina y Zootecnia de Rumiantes, Facultad de Medicina Veterinaria y Zootecnia, Universidad Nacional Autónoma de México, Mexico City 04510, Mexico; delaluzarmendarizj@fmvz.unam.mx (J.D.l.L.-A.); alberti@unam.mx (A.B.A.-N.); ginahernandezrojas@fmvz.unam.mx (E.G.H.-R.); ducoing@fmvz.unam.mx (A.E.D.-W.); 2Programa de Doctorado en Ciencias de la Producción y de la Salud Animal, Universidad Nacional Autónoma de México, Mexico City 04510, Mexico; aljogaba@gmail.com; 3Laboratorio de Virología, Centro Nacional de Investigación Disciplinaria en Salud Animal e Inocuidad, Instituto Nacional de Investigaciones Forestales, Agrícolas y Pecuarias (INIFAP), Mexico City 04010, Mexico

**Keywords:** small ruminant lentiviruses, goat, sheep, Mexico, genotypes

## Abstract

Small ruminant lentivirus is widely distributed in the world, and its infection causes production and economic losses for goat and sheep producers. In this study, the presence of this virus was determined in goat and sheep flocks in the northern, central, and southern regions of Mexico at a frequency of 74 to 82%, with the central region having the highest identified frequency. Furthermore, genotype B was confirmed to be the most frequent genotype in the goat and cohabitation flocks, while genotype A was identified more frequently in the sheep flocks and was not detected in the goat-only flocks. The viruses circulating in Mexico were confirmed to mainly cause respiratory signs in sheep and to cause mastitis and arthritis in goats. Finally, regarding production in Mexico, the main risk factor for the circulation and dissemination of this virus is the cohabitation of goats with sheep.

## 1. Introduction

Small ruminant lentiviruses (SRLVs) cause persistent, chronic degenerative, multisystem diseases in goats and sheep [1,2,3]. Clinical signs are associated with the respiratory tract, nervous system, musculoskeletal system, and mammary glands. These diseases are characterized by inflammatory lesions in various organs, with the most affected being the lungs, brain, mammary glands, and joints. Sheep are usually the most susceptible species, with clinical pictures that include progressive interstitial pneumonia leading to dyspnea and weight loss, and demyelinating leukoencephalomyelitis leading to neurological signs, indurative mastitis, and arthritis. In contrast, young infected goats usually present with leukoencephalitis, while chronic arthritis and mastitis are more common in adults. These conditions result in prolonged illness and reduced productivity, exerting a significant negative economic impact on the small ruminant industry. The incubation period typically lasts 6 months after infection, with approximately 35% of animals exhibiting clinical signs [4,5].

Colostrum and milk have been identified as the main, but not the only, routes of transmission of SRLV [6]. In practice, control measures and eradication programs focus mainly on separating the offspring from their mothers at birth and feeding them colostrum and milk obtained from SRLV-free animals or heat-treated colostrum [7,8]. Other transmission routes include direct and prolonged contact with infected animals, particularly in sheep that develop ovine progressive pneumonia (OPP), where respiratory exudates with high viral loads can facilitate transmission of the virus [9]. This mode of transmission is more effective in crowded conditions or where high densities of stabled or grazing sheep are kept [6]. Some studies have indicated the importance of venereal or mother-to-offspring transmission [10]; however, other authors argue that there is no epidemiological relevance [11]. In dairy goats, SRLV infection leads to an approximately 6% decrease in milk production, while in sheep, it alters milk characteristics, specifically lactose and fat levels [12]. In goats raised for meat production, the infection reduces daily weight gain, with lambs showing a loss of 940 g of meat, alongside a decrease in overall daily weight gain [13,14,15].

Historically, SRLVs are known as two different viral agents, one being the caprine arthritis encephalitis virus (CAEV) infecting goats and the other being the Maedi Visna virus (MVV) infecting sheep [16,17]. The MVV was the first lentivirus to be identified in sheep in Iceland in the 1950s [13]. In the 1970s, the CAEV was identified from goats [2]. Both agents have been found in both species, so they are considered part of the SRLV group due to their ability to cross the interspecies barrier [18].

Small ruminant lentiviruses (SRLVs) are single-stranded RNA viruses grouped in the order *Ortervirales*, the family *Retroviridae*, the subfamily *Orthoretrovirinae*, and the genus *Lentivirus* [19]. SRLVs are classified into five genogroups (A–E). Genotype A consists of MVV-like strains, with 22 subtypes (A1–A22), and genotype B corresponds to CAEV-like isolates, with 5 subtypes (B1–B5). Genotype E is divided into subtypes E1 and E2, which have been identified exclusively in goats from Italy [20,21,22,23].

Globally, the prevalence of SRLV can reach up to 90%, with risk factors for infection spread including the productive stage, age, breed, and production purpose, as well as facility-related factors such as cohabitation with other species, population density, preventive measures, cleanliness, biosecurity, and management practices [24,25,26,27].

The epidemiology of SRLVs represents a difficult and complex problem, as it involves a wide range of biosecurity and diagnostic aspects. In many countries, goat and sheep herds are often raised in low-density flocks and on family farms. These two aspects can strongly influence sanitary management, increasing the risks of SRLV transmission and reducing biosecurity levels within farms. Accurate data on the probability of exposure and transmission, as well as data on surveillance and identification of circulating viral genotypes, are still scarce. Although goat and sheep farming is a smaller sector (in economic terms) than other livestock activities, the evaluation of the factors that influence SRLV transmission can considerably improve animal health, welfare, and production programs [28].

In Mexico, the primary goat- and sheep-producing regions are located in the central and southern parts of the country, with the reported prevalence of SRLV reaching 35% [18]. Genetic characterization studies have revealed that genotypes A and B circulate naturally within these production units. This study was performed to explore the genetic diversity of SRLV in Mexico and to determine the risk factors that promote its presence in sheep and goats across different flocks.

## 2. Materials and Methods

### 2.1. Study Population and Sample Collection

This study analyzed a total population of 890 goats (*n* = 383) and sheep (*n* = 507) from 52 cohabitation and individual flocks located in the northern (Aguascalientes, Zacatecas, Nayarit, Sinaloa, and Jalisco), central (Querétaro, Estado de México, Guanajuato, Ciudad de México, Morelos, Puebla, and Hidalgo), and southern (Veracruz, Oaxaca, and Guerrero) regions of Mexico. The inclusion criterion was the observation of clinical signs associated with viral infection. The sample size was estimated using the formula *n* = (z2 × p × q)/d2 with a confidence level of 99% and an expected prevalence of 50%. Based on this calculation, 11 individuals were sampled from each production unit, totaling 507 sheep and 383 goats.

Peripheral blood samples were collected via jugular venipuncture into tubes without an anticoagulant for serum collection and with an anticoagulant for peripheral blood mononuclear cell collection using gradient separation with Ficoll Paque Plus^®^ (GE Healthcare Bio-Sciences, Pittsburgh, PA, USA). Both sample types were stored at −70 °C until processing. Additionally, a structured questionnaire with closed questions was administered in each production unit to gather information related to SRLV presence, transmission, and dissemination. The questionnaire covered variables such as clinical signs, preventive management (e.g., deworming, vitamin administration, and biosecurity practices), corral cleaning, and type of production unit.

For the analysis of risk factors, the following variables were included: species, breed, region, zootechnical function, type of production system, age, herd size, type of herd, technical training of the producer, knowledge of these diseases, participation in livestock fairs or exhibitions, contact with other herds, introduction of replacements, use of artificial insemination, mounting with external stallions, separation or isolation of sick animals, and veterinary assistance. Table 1 shows the numbers and types of flocks by region.

### 2.2. Serological Diagnosis

In total, 890 serum samples were analyzed using the Small Ruminant Lentivirus Antibody Test Kit, cELISA (Veterinary Medical Research & Development, Pullman, WA, USA), a competitive immunoenzymatic assay. The manufacturer’s instructions were followed for the procedure and validation. This test detects antibodies specific for SRLV glycoprotein 135, recognizing genotypes A and B with a sensitivity of 95% to 100% and a specificity of 98.4% to 99.6% [29,30].

### 2.3. Molecular Diagnosis

DNA extraction was performed on the peripheral blood mononuclear cell samples (*n* = 890) using the DNeasy Blood & Tissue Kit (Qiagen, Hilden, Germany). Endpoint polymerase chain reaction (PCR) was then conducted to amplify a 233 bp fragment of the non-coding region of the virus (long terminal repeat), using primers described by Sánchez et al. [31]. The PCR was carried out using the GoTaq Green Master Mix^®^ (PROMEGA, Madison, WI, USA) with the following concentrations: 12.5 µL of Master Mix Green, 2 µL of the forward primer, 2 µL of the reverse primer, 3.5 µL of nuclease-free water, and 5 µL of the DNA. The thermocycler program was as follows: initial denaturation at 94 °C for 5 min, followed by 35 cycles of denaturation at 94 °C for 30 s, annealing at 55 °C for 44 s, and extension at 72 °C for 30 s, and a final extension at 72 °C for 10 min. The PCR product was visualized through horizontal electrophoresis on a 2% agarose gel stained with ethidium bromide.

### 2.4. Genotyping of Positive Samples

Samples that tested positive in the molecular diagnostic test (*n* = 697) were further analyzed to identify the corresponding genotype (A/B) using real-time PCR. The primers used were previously reported by Kuhar et al. [32]. The commercial LightCycler^®^ SYBR Green I Master Kit (Roche Diagnostics, Mannheim, Germany) was used, with the reaction mixture consisting of 5 µL of Master Mix, 0.75 µL of the forward primer, 0.75 µL of the reverse primer, 2.5 µL of water, and 2 µL of the DNA. The thermocycler program was as follows: initial denaturation at 95 °C for 3 min, followed by 40 cycles of 95 °C for 15 s, and a final cycle at 60 °C for 30 s. A melting curve was also generated, with an initial cycle at 65 °C for 30 s followed by 60 cycles at 65 °C for 5 s, with the temperature increasing by 0.5 °C per cycle.

### 2.5. Risk Factors Associated with the Presence of SRLV

The information obtained from the surveys of the 52 flocks was analyzed in relation to SRLV molecular detection, genotyping, seropositivity, and possible risk factors. The possible risk factors analyzed were age, breed, herd size, mixed herd production, technical training of the producer, producer knowledge of SRLV diseases, participation in livestock fairs, contact with other herds, introduction of replacements from other herds, use of artificial insemination, mating with stallions, isolation of sick animals, and veterinary assistance in the herds. The variables corresponding to the questionnaires were grouped into clusters: species (goat or sheep), geographic region (central, northern, and southern), food production system (milk and meat), and production system (intensive and semi-intensive).

To analyze the clusters, 2 × 2 contingency tables were generated, and the chi-square test was employed with a significance level of α = 0.05 to assess the independence of the variables. For factors that demonstrated a statistically significant association (*p* < 0.05), odds ratios (ORs) and 95% confidence intervals (CIs) were calculated. Factors found to be independent of the presence of SRLV or antibodies in any of the analyzed strata were excluded from further analysis.

The flocks showing dependence in the presence of the risk factors were used as the analysis group for calculating the OR value. Identifying these groups allowed for quantitative comparisons between the study groups, enhancing the interpretation of the magnitude and direction of the observed associations. This analytical approach ensured that the calculated OR accurately reflected the relative probability of the event of interest, thereby providing a robust basis for interpreting the results. Statistical analyses were performed using R (version 4.3.1) via RStudio (version 2023.06.0), utilizing the “epibasix” package for elementary epidemiological and biostatistical functions.

To identify the effect of management conditions, type of production (simple or cohabitation), in flocks with clinical signs (respiratory signs, nervous disorders, arthritis, and mastitis) with a population of goats or sheep negative in the laboratory tests performed, the results were analyzed using contingency tables and a logistic regression model. The flocks included in this analysis were four goat flocks, seven sheep flocks and six cohabiting flocks (mixed flocks). The data analyses were conducted using the JMP statistical software package, version 14.3.0 (SAS Institute, Cary, NC, USA).

## 3. Results

### 3.1. Seroposivity and Molecular Detection of SRLV-LTR

Of the 890 animals sampled (507 sheep and 383 goats), 40% (354/890) tested positive for specific antibodies to SRLV and 78% (697/890) tested positive for the presence of the viral genome (LTR). Among the goats, 60% (229/383) were seropositive and 89% (340/383) were positive for the molecular detection of SRLV. By contrast, 25% (125/507) of sheep had specific antibodies, and 70% (357/507) had the viral genome.

In the northern region of Mexico, 40% (111/279) of goats and sheep were seropositive, and 76% (213/279) tested positive for the viral genome. Within this region, goat flocks had a higher number of positive animals, with 71% seropositive and 94% (94/100) positive for molecular detection. In the southern region, the frequency of seropositive goats and sheep was 45% (186/411), and the rate of molecular detection was 82% (337/411). The highest seropositivity (76%) was observed in the goat-only flocks, and the molecular detection rates were similar in both the mixed-species and goat-only flocks (93%). The central region recorded 57/200 seropositive animals (28%) and 147/200 animals positive for the viral genome (74%). The highest frequency of seropositive animals (37%) and animals with positive viral genomes (85%) in this region was found in the goat-only flocks. Figure 1 shows the frequency of viral presence in each region.

### 3.2. SRLV Genotyping

Across the northern, central, and southern regions of Mexico, the circulation of SRLV genotype A was confirmed in 15% (105/697) of cases, genotype B in 66% (458/697), and co-infections in 19% (134/697). Genotype A was identified in 11% (33/308) of mixed-species flocks and in 37% (72/195) of sheep, with the central region having the highest percentage of positive animals (21%). Genotype B and co-infections were identified in all three types of flocks across the three regions, with genotype B being most frequent in the southern region (78%) and co-infections most common in the central region (21%). Figure 2 shows the percentage frequency of SRLV genotypes by region.

### 3.3. Risk Factors

Several factors were identified that may influence the presence of antibodies or the molecular identification of SRLV in the three geographic regions analyzed in Mexico (Appendix A). The risk factors were analyzed using data from questionnaires administered to producers and other data collected during visits to flocks.

The risk factor analysis showed a significant association (*p* < 0.05) between SRLV seropositivity and age, flock size, and veterinary assistance in some clusters. When determining LTR positivity via PCR, production in mixed flocks was added as a variable in the cluster for the central region, as were knowledge of SRLV diseases and contact with other flocks in the sheep cluster. In the identification of genotypes A or B, the variables that were significant, in addition to those mentioned, were technical training by the producer, participation in livestock fairs, isolation of sick animals, and veterinary assistance.

The variables that did not present a significant association (*p* > 0.05) were breed, introduction of replacements, use of artificial insemination, and mating with a stallion from another flock. Likewise, the clusters made up of animals destined for milk production were not associated with any risk factor.

The goat flocks had a significantly higher frequency of viral positivity (*p* < 0.001) than the mixed-species (89.1%) and sheep-only (88.2%) flocks. The frequency of infection in the goats (94.5%) was significantly higher (*p* < 0.001) than that in the sheep (87.9%). Additionally, the production system significantly influenced the positivity rates (*p* < 0.001), with intensive systems showing a 93% positivity rate compared with only 39% in the semi-intensive systems.

In analyzing the association between clinical signs and infection frequency, a significant association was found between positivity and respiratory signs in sheep (*p* < 0.001), with higher rates observed in the mixed-species and sheep-only flocks than in the goat-only flocks. Clinical signs pertaining to the nervous system were observed in only 5.76% of the sampled sheep, with no significant association between these signs and infection frequency (*p* > 0.05). Mastitis was observed in 32% of cases and was significantly associated (*p* < 0.001) with infections in goats, with no cases detected in sheep. Similarly, arthritis was observed in 42.6% of goats and was significantly associated (*p* < 0.001) with the frequency of infections in this species.

## 4. Discussion

The results of this study show that 51% (354/697) of goats and sheep were positive for the presence of the SRLV genome but negative for specific antibodies. These findings suggest that, in Mexico, molecular diagnostic tests should be recommended as a confirmatory tool to accurately identify animals infected with SRLV. Czopowicz et al. [33] reported similar results when comparing both tests and attributed the low sensitivity of the enzyme-linked immunosorbent assay (ELISA) tests to factors such as late seroconversion, fluctuating antibody titers during SRLV infection, or the high variability of circulating genotypes. The serological tests used in Mexico are imported and are based on strains different from those circulating locally. This discrepancy, combined with the high genetic variability of SRLV, likely contributes to a high number of false-negative results. Herrmann-Hoesing et al. [34] determined that the sensitivity and specificity of ELISA tests are influenced by the strain used in the test, emphasizing the need for tests developed using locally circulating strains to ensure accurate detection. These findings underscore the importance of using both serological and molecular tests to develop effective prevention and control strategies in production units.

Regarding the identified risk factors, positivity based on serology or PCR occurs mostly in goats older than two years of age, which are found in the central region of the country, regardless of the type of production system (intensive or semi-intensive). The size of the flock (>50 animals) represented a risk factor because the majority of the analyzed flocks (50/52) had a population of less than 50 goats or sheep. The production of mixed flocks had an OR of 3.31 for the central region. When performing genotyping, the infection was identified to occur in the goat-to-sheep direction, since genotype B was present in sheep and genotype A was not identified in goats. This result shows the importance of seeking single-species production systems to avoid between-species SRLV infections. Jacob-Ferreira et al. [35] found in a seroprevalence study on the association of risk factors in mixed herds in northern Portugal that risk factors such as species (goats), age (over two years), herd size (>100), intensive production system, and participation in livestock competitions, among others, represent risk factors that increase the probability of being seropositive.

An important point to note is that even producers having knowledge of these diseases or technical training has not been sufficient to prevent the spread of SRLV, so reinforcements of this control strategy or the proposal of a different solution is important.

Shah et al. [22] reclassified SRLVs, identifying clade A, which has a tropism for sheep, and clade B, which is more commonly associated with goats, with a genetic homology between 75% and 85%. Pisoni et al. [36] further reported that genotypes A and B do not have strict host tropism, meaning that the cohabitation of goats and sheep increases the likelihood of both species being infected by either genotype.

In the present study, SRLV genotype B was detected in the goat, sheep, and mixed-species production units. The proportion of positive animals was higher in the goat flocks than in the sheep-only or mixed-species flocks, with 91% of animals in the mixed-species flocks testing positive for genotype B. This indicates that genotype B circulates more widely among goats and that sheep cohabiting with infected goats are at greater risk of infection.

In Brazil, De Azevedo et al. [37] conducted a genetic study that similarly showed genotype B to be more prevalent than genotype A in goat production units. Genotype A was more frequently detected in sheep production units, while mixed-species units showed lower frequencies of animals positive for both genotypes. These findings support those of L’Homme et al. [38], who concluded that genotype A, when transmitted from sheep to goats, undergoes genetic modifications that enhance its ability to cause diseases. Kuhar et al. [32] also confirmed the emergence of subtype A14 in goats that cohabitate with sheep, suggesting that such cohabitation not only predisposes both species to infection with genotypes A and B but also may facilitate the emergence of new genotypes.

Respiratory clinical signs such as coughing, nasal discharge, and sneezing were associated with SRLV infection in both the mixed-species flocks and sheep-only flocks. These findings are consistent with a previous report by Luján et al. [39], who described SRLV-induced interstitial pneumonia in adult sheep, noting that respiratory disorders are less common in goat kids. The respiratory route is a critical pathway for the excretion and dissemination of the virus.

Hasegawa et al. [40] reported that arthritis and indurative mastitis are clinical signs characteristic of goats infected with SRLV. Juste et al. [41] further confirmed that SRLV infection leads to economic and production losses in Latxa dairy flocks, estimating a loss of approximately 13 L of milk per year per infected ewe, which equates to an economic loss of approximately 12–24 € per ewe annually between 1999 and 2010. In this study, arthritis and mastitis were observed in adult goats within production units, with these clinical signs significantly associated with SRLV presence, likely contributing to productivity losses in goat production in Mexico.

Studies from European countries such as Poland, Belgium, and Italy have concluded that a high degree of specialization in dairy production and large production units are major predisposing factors for lentivirus infection in small ruminants [26,42,43,44]. These results are consistent with our findings, which demonstrate that intensive production units with larger numbers of animals were significantly more prone to SRLV infections. However, in our study, the highest number of positive animals was found in flocks lacking biosecurity, cleaning, and preventive measures. This highlights the importance of implementing biosecurity and prevention strategies in all production units to prevent the introduction, establishment, and spread of viral agents in goats and sheep in Mexico.

## 5. Conclusions

In conclusion, these results demonstrate that SRLV genotype A circulates in sheep production units, mainly causing clinical respiratory signs. Genotype B occurs more frequently in goat production units associated with mastitis and arthritis. Finally, the production units in which goats cohabitate with sheep predispose both species to infections with the two genotypes. Therefore, the implementation of biosafety strategies is extremely important as part of preventive medicine programs in sheep and goat production in Mexico.

## Figures and Tables

**Figure 1 vetsci-12-00204-f001:**
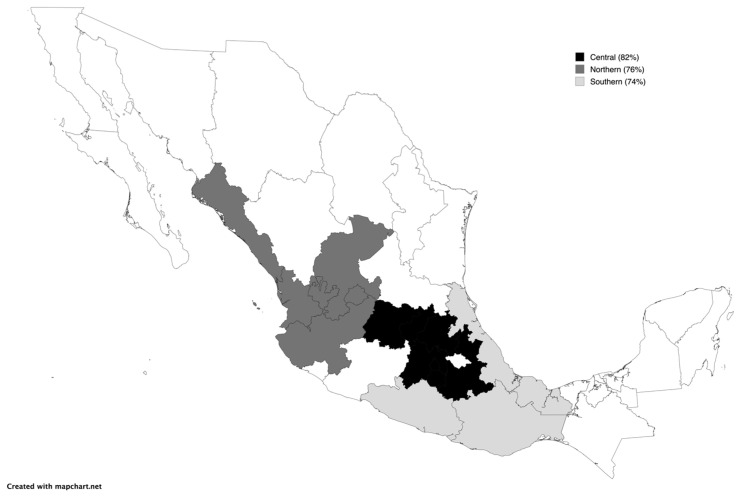
Distribution of molecular positivity of SRLV (PCR-LTR) identified by region in Mexico.

**Figure 2 vetsci-12-00204-f002:**
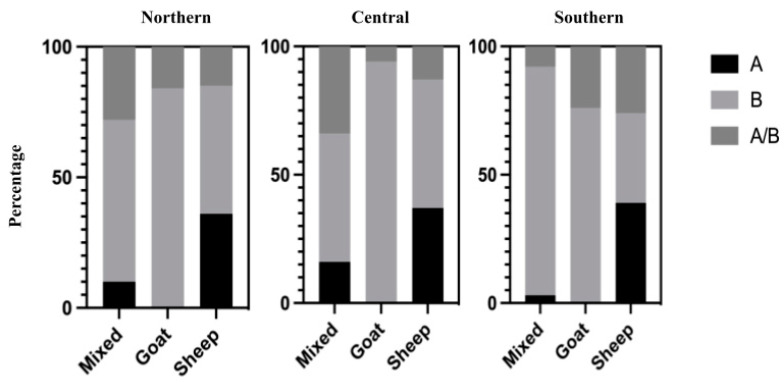
Frequency (percentage) of SRLV infections with genotypes A, B, or both identified by region.

**Table 1 vetsci-12-00204-t001:** Distribution and type of production units by region in this study.

Region	Mixed (Goat/Sheep)	Goat	Sheep	Total
North	7 (53/65)	3 (100)	4 (61)	14 (153/126)
Center	7 (59/92)	2 (71)	16 (189)	25 (130/281)
South	5 (60/60)	4 (40)	4 (40)	13 (100/100)
Total	19 (172/217)	9 (211)	24 (290)	52 (383/507)

## Data Availability

Data are contained within the article and Appendix A.

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
