# Peer review of "Distribution of Small Ruminant Lentivirus Genotypes A and B in Goat and Sheep Production Units in Mexico"

_vetsci, 2025, doi:10.3390/vetsci12030204_

Round 1
Reviewer 1 Report
Comments and Suggestions for Authors
After searching the literature, I noticed that the findings presented in this manuscript are not new. The authors would have gone beyond just the broad patterns of SRLV genotype distribution and clinical associations in Mexico. Although the results presented are consistent with previous studies. The authors could have focused on the analysis of regional and management-related factors, providing a unique contribution to the literature. To strengthen the manuscript's novelty, the authors could:
- Expand on potential evolutionary or ecological drivers for the observed patterns in Mexico.
- Compare their findings with those from other countries to highlight regional differences.
- Explore emerging trends, such as genetic shifts in SRLV due to mixed-species cohabitation or management practices unique to the region.
This would position the study as both a regional analysis and a contributor to the global understanding of SRLV dynamics.
Comments on the Quality of English LanguageSeveral sections contain minor grammatical errors and awkward phrasing
Reviewer 2 Report
Comments and Suggestions for Authors
Small ruminant lentivirus is widely spread in the world and causes productive and economic losses. The epidemiological investigation is useful for better control of the disease. The author determined the presence and genotype distribution of the virus in the northern, central and southern areas of Mexico, and analyzed the relationship of clinical signs and factors with the presence of different SRLV genotypes.
Overall, the study is too simple and the results is not shown in detail.
1.This study focus on genotype A and B SRLV, the virus name should also be clarified in the Introduction part.
2.What’s the differences and improvement of this study with previous publication(ref 18)?
3.The author could supplement positive rate data in Figure 1.
4.The author could determine the subtypes of the detected SRLVs.
5.The author could perform phylogenetic analysis of the detected SRLVs.
6.Associated factors analysis should be supported by figures or tables.
7.The abbreviation of Small ruminant lentiviruses should be shown in the first line(line 25).
8.line 72, the clinical signs associated with viral infection should be clarified here.
Round 2
Reviewer 1 Report
Comments and Suggestions for Authors
The current version of the manuscript demonstrates significant improvements compared to the previous one. The authors have carefully addressed the feedback provided, making substantial revisions and enhancements to key sections.
Author Response
Comment: The current version of the manuscript demonstrates significant improvements compared to the previous one. The authors have carefully addressed the feedback provided, making substantial revisions and enhancements to key sections.
Response: Many thanks to the reviewer for making suggestions that have led to an improved manuscript.